

# Genome-wide identification and characterization of Dof transcription factors in eggplant (*Solanum melongena* L.)

Qingzhen Wei,  Wuhong Wang,  Tianhua Hu,  Haijiao Hu,  Weihai Mao,
Qinmei Zhu and  Chonglai Bao

Institute of Vegetable Research, Zhejiang Academy of Agricultrual Sciences, Hangzhou, Zhejiang, China

## ABSTRACT

Eggplant (*Solanum melongena* L.) is an important vegetable cultivated in Asia, Africa and southern Europe and, following tomato and pepper, ranks as the third most important solanaceous vegetable crop. The *Dof* (DNA-binding with one finger) family is a group of plant-specific transcription factors that play important roles in plant growth, development, and response to biotic and abiotic stresses. The genes in the Dof family have been identified and analysed in many plant species, but the information remains lacking for eggplant. In the present study, we identified 29 *SmeDof* members from the eggplant genome database, which were classifed into nine subgroups. The phylogeny, gene structure, conserved motifs and homologous genes of *SmeDof* genes were comprehensively investigated. Subsequently, we analysed the expression patterns of *SmeDof* genes in six different eggplant subspecies. The results provide novel insights into the family of *SmeDof* genes and will promote the understanding of the structure and function of *Dof* genes in eggplant, and the role of *Dof* expression during stress.

Corresponding author
Chonglai Bao, baocl@mail.zaas.ac.cn

# INTRODUCTION

Eggplant (*Solanum melongena*) is an economically important vegetable crop in the large family Solanaceae, which contains other widespread crops including tobacco, potato, tomato and pepper. Endemic to the Old World, eggplant was domesticated in the region of India and southeast China and is now cultivated worldwide (*Doganlar et al., 2002*). However, eggplant is susceptible to many abiotic stresses and diseases such as *Fusarium* wilt (caused by *Fusarium oxysporum*) and *Verticillium* wilt (caused by *Verticillium dahlia*), thus pose a substantial threat to production and food security. The availability of genome sequences and development of bioinformatics approaches provide a good opportunity to characterize a family of stress resistance-related genes from a whole-genome perspective.

The Dof (DNA-binding with one finger) proteins are a set of plant-specific transcription factors (TFs) that typically have 200–400 amino acids and two major domains: a C-terminal domain for transcriptional activation and a highly conserved N-terminal DNA-binding domain (also referred to as the Dof domain).The Dof domain is composed of 52 amino

acid residues and features a C2C2-type zinc finger motif, some of which could recognize the specific regulatory elements of AAAG or CTTT in the promoters of target genes (*Plesch, Ehrhardt & Mueller-Roeber, 2001*; *Wray et al., 2003*; *Yanagisawa, 2004*; *Chen et al., 2014*; *Mehrotra et al., 2014*), although the pumpkin Dof protein AOBP recognizes an AGTA repeat (*Shimofurutani et al., 1998*). Similar to other zinc fingers, the Dof domain plays a bi-functional role in DNA-binding and protein-protein interactions (*Yanagisawa, 2002*). The Dof family may have evolved from a common ancestor: a single gene in the green unicellular alga *Chlamydomonas reinhardtii*, with the family then expanding into mosses, ferns and vascular plants by consecutive gene duplications (*Moreno-Risueno et al., 2007*). *Yanagisawa (2002)* first identified 37 putative *Dof* genes in *Arabidopsis*, which were classified into eight groups. *Lijavetzky, Carbonero & Vicente-Carbajosa (2003)* classified *Dof* genes into classes A, B1, B2, C1, C2.1, C2.2, C3, D1 and D2, which is the classification system adopted in the analysis of *Dof* genes of Chinese cabbage (*Ma et al., 2015*) and carrot (*Huang et al., 2016*) or modified such as in tomato (*Cai et al., 2013*).

Dof transcription factors are associated with a variety of biological processes in plant growth and development, including the responses to light and those in defence (*Park et al., 2003*; *Shaw et al., 2009*; *Zhang et al., 1995*), germination and development of seeds (*Papi et al., 2002*; *Dong et al., 2007*; *Gabriele et al., 2010*; *Zou et al., 2013*), control of flowering (*Imaizumi et al., 2005*; *Fornara et al., 2009*), branching of shoots, and development of pollen, fruit and vascular tissues (*Konishi & Yanagisawa, 2007*; *Chen et al., 2012*; *Da Silva et al., 2016*; *Wu et al., 2016*). Dof TFs also participate in carbon metabolism (*Yanagisawa, 2000*) and various physiological processes such as nitrogen assimilation, phytochrome and cytochrome signalling, plant hormonal signalling, and biotic and abiotic stress tolerance (*Park et al., 2003*; *Ward et al., 2005*; *Kurai et al., 2011*; *Cai et al., 2013*; *Corrales et al., 2014*; *Wen et al., 2016*). For example, the newly found Dof protein MaDof23 in banana functions as a transcriptional repressor and interacts with the transcriptional activator MaERF29. The two proteins are suggested to control banana fruit ripening by working antagonistically to regulate 10 ripening-related genes involved in cell wall degradation and aroma formation (*Feng et al., 2016*). In *Arabidopsis*, Dof6 acts as an opponent of TCP14, which is a positive regulator of seed germination, to regulate ABA biosynthesis pathway-related genes, and therefore, seed germination is negatively affected (*Rueda-Romero et al., 2012*).

Dof TFs have been functionally characterized in many plant species and crops such as *Arabidopsis*, tomato, rice, cucumber and soybean (*Papi et al., 2000*; *Gualberti et al., 2002*; *Washio, 2003*; *Cai et al., 2013*; *Guo & Qiu, 2013*; *Wen et al., 2016*). However, information on *Dof* genes continues to be lacking for eggplant. The aim of the present study was to conduct a genome-wide analysis of the *Dof* family of genes in eggplant. We identified 29 *SmeDof* members in the eggplant genome and classified these proteins into nine subgroups. The physical and chemical characteristics, gene structure, motifs and evolutionary patterns of the *SmeDof* family of genes were investigated. We also analysed the expression patterns of *SmeDof* members in different types of eggplant. Based on this study, our understanding of the structure and function of the *SmeDof* family of genes in eggplants has increased, and the results will contribute to futher understanding the role of *SmeDof* genes in tolerance mechanism and the interaction networks of these genes.

## MATERIALS AND METHODS

### Identification and characterization of the Dof transcription factors

The genome, genes and corresponding protein sequences of eggplant were downloaded from the Eggplant Genome Database (http://eggplant.kazusa.or.jp). The latest Markov model for the Dof transcription factors named PF02701.13 was downloaded from the Pfam database (http://pfam.xfam.org/) (*Eddy, 2011*; *Finn et al., 2014*). The HMMER program was used to search for Dof proteins in all eggplant proteins with a cutoff $E$-value of $1e^{-4}$ using PF02701.13 as a query. After a comprehensive check, the candidate proteins that only contained fragmental Dof domains were eliminated.

The Dof transcription factor search was also conducted for five other representative species in the plant kingdom and for the family Solanaceae. The annotated proteins of an alga (*C. reinhardtii*), moss ( *Physcomitrella patens*), lycophyte (*Selaginella moellendorffii*), *Arabidopsis thaliana*, grape (*Vitis vinifera*), poplar (*Populus trichocarpa*) and rice (*Oryza sativa*) were downloaded from the Pfam database (v10) (*Goodstein et al., 2012*). The genomic information of the gymnosperm Norway spruce (*Picea abies*) (Nystedt et al., 2013) was collected from the Congenie Website (http://congenie.org/). The annotated proteins of petunia (*Petunia axillaris*), tomato (*Solanum lycopersicum*), potato (*Solanum tuberosum*), pepper (*Capsicum annuum*), and tobacco (*Nicotiana tabacum*) were downloaded from the SGN Database FTP Server (ftp://ftp.solgenomics.net/genomes/).

### Phylogenetic analysis of the *Dof* genes

The Dof proteins of *C. reinhardtii*, *P. patens*, *A. thaliana*, *O. sativa* and eggplant were selected for phylogenetic analysis in *planta*. The ClustalX2 program was used to align Dof protein sequences with the Gonnet protein weight matrix (*Larkin et al., 2007*). The MEGA program (v6.06) was used to construct a neighbour-joining phylogenetic tree using the Jones-Taylor-Thornton (JTT) model with 500 bootstrap replicates (*Tamura et al., 2013*). The uniform rates and homogeneous lineages were adopted, whereas the partial deletion with a site coverage cutoff of 70% was used for gaps/missing data treatment. To clearly distinguish the genes in this phylogenetic tree, the term 'Nta|' was added as a prefix to indicate the genes were from tobacco. The MEGA program was also used to construct a neighbour-joining phylogenetic tree of Dof proteins in plants of Solanaceae, following previous methods. The frequency of each divergent branch was displayed when the value was higher than 50%. Adobe Illustrator software was used to clearly show the Dof branches after classification of all proteins based on the known background information.

### Gene structure and motif analyses

The Gene Structure Display Server tool (http://gsds.cbi.pku.edu.cn/, v2.0) (*Hu et al., 2015*) was used to analyse gene structure. MEME software (http://meme-suite.org/meme_4.11.0/, v4.11.0) was used to search for motifs in all 29 eggplant Dof proteins with a motif window length from 8 to 100 bp (*Bailey et al., 2009*). The default number of motifs to be found was set to 15, resulting in identification of 14 motifs with an $E$-value smaller than the 0.05 significance level. The motifs were displayed following a pattern with the most statistically significant (lowest $E$-value) motifs displayed first.

## Identification of orthologous and paralogous genes

OrthoMCL (v2.0.3) (*Li, Stoeckert Jr & Roos, 2003*) was used to search for orthologous, co-orthologous and paralogous genes in eggplant, tomato, *Arabidopsis*, and rice using the entire Dof protein sequence data set. The *E*-value cutoff of $1.0e^{-5}$ was used for the all-against-all BLASTP alignment process, and a match cutoff value of 50 was adopted. The orthologous and paralogous relationships were gathered and displayed using Cytoscape software (http://www.cytoscape.org, v2.8.3) (*Smoot et al., 2011*).

## Analysis of *Dof* gene expression in different eggplants

The genome-wide digital gene expression of *SmeDof* genes in eggplant varieties and close species was assessed as described by *Tan et al. (2015)*. We used RNA-seq transcriptiomes with the related FPKM (fragments per kilobase per million measure) values to study *SmeDof* gene expression patterns in six eggplants. Illumina sequencing reads from RNA-seq data of six types of eggplants with different morphological traits were used (raw data is avalible at NCBI, SRP127743), including two cultivated eggplants *Solanum melongena* L. (MEL_HZHQ and MEL_LYQ), three closest wild relatives *S. Aethiopicum, S. sisymbriifolium* and *S. integrifolium* (AET_SG, SIS_2007 and INT_2006) and a semi-wild relative *S. melongena* (MEL_S58). Seeds (10 per species) from six types of eggplant were germinated. The leaf tissue was collected from the seedlings with three true leaves when 14 days post germination. Total RNA of each sample was extracted using an RNAprep Pure Plant Kit (DP432, TIANGEN, China. http://www.tiangene.com) according to the manufacturer's protocol. Illumina sequencing was performed at Novogene Bioinformatics Technology Co., Ltd., Beijing, China (http://www.novogene.cn). The library preparations were sequenced on an Illumina Hiseq 2000 platform and 90 bp paired-end reads were generated. Approximately 28 million paired reads were generated with 5.6 G data for each sample. The program Tophat (https://ccb.jhu.edu/software/tophat/) was used to map the reads to the eggplant genome (*Hirakawa et al., 2014*); then the expression profile of all spinach genes was obtained with FPKM (Fragments Per Kilobase of exon per million fragments Mapped) value using software Cufflinks (http://cole-trapnell-lab.github.io/cufflinks, v2.2.1) under the guidance of annotated gene models with a GFF file. The *SmeDof* gene expression profile from each sample was analysed using the HemI program (http://hemi.biocuckoo.org/) with the average hierarchical clustering method.

## RESULTS AND DISCUSSION

### Genome-wide identification of *Dof* genes in eggplant

Gene family analysis is an efficient approach to understand the structure, function, and evolution of genes. The *Dof* genes are plant-specifc transcription factors that are ubiquitous in plant species and participate in various biological processes. To identify *SmeDof* homologues in eggplant, we used the HHM profile of the Dof domain (PF02701.13) as a query to perform an HMMER search against the Eggplant Genome Database (http://eggplant.kazusa.or.jp). A total of 85,446 genes were predicted in eggplant genome (versione SME_r2.5.1), including 42,035 predicted genes (38,498 intrinsic genes and 3,537 partial genes), 41,048 transposable elements and 2,363 were pseudo or short genes or both.

A total of 29 genes encoding transcription factors of the *SmeDof* family were identified. The number is similar to that in rice (30 *OSDof* genes; *Washio, 2003*) but lower than the number of *Dof* genes in *Arabidopsis* (36 *At Dof* genes; *Lijavetzky, Carbonero & Vicente-Carbajosa, 2003*), soybean (78 *GmDof* genes; *Guo & Qiu, 2013*), Chinese cabbage (76 *BraDof* genes; *Ma et al., 2015*), potato (35 *StDof* genes; *Venkatesh & Park, 2015*), cucumber (36 *CsDof* genes; *Wen et al., 2016*), and tomato (34 *SlDof* genes; *Cai et al., 2013*). For convenience, the 29 *SmeDof* genes were assigned names from *SmeDof-01* to *SmeDof-29*.

The 29 *SmeDof* genes in eggplant were classified into four groups and nine subgroups based on predicted Dof domains: A, B1, B2, C1, C2.1, C2.2, C3, D1 and D2 (Fig. 1, Table 1). In a previous study on Dof transcription factors from green unicellular algae to vascular plants, *Dof* genes are classified into groups A, B, C, D, E, F and G (*Moreno-Risueno et al., 2007*), whereas group G is absent from *Dof* gene analysis between sorghum and rice (*Kushwaha et al., 2011*). The classification of the 29 *SmeDof* genes was consistent with that of *Arabidopsis*, although the number of members in each subgroup might vary. However, in cucumber, the *CsDof* genes were classified into eight subgroups, with no genes found in subgroup C3 (*Wen et al., 2016*). The genome-wide analysis of the Dof transcription factors classified the soybean *Dof* genes into nine different groups: A, B1, B2, C1, C2, C3, D1, D2, and D3 (*Guo & Qiu, 2013*).

The *SmeDof* genes had molecular weights ranging from 18.6 to 64.4 kD with pI values varying from 4.55 to 9.48. Two eggplant genes *Sme2.5_00232.1_g00011* and *Sme2.5_01519.1_g00001* were identified as candidates with the *E*-value cutoff of 2.2e–09 and 9.7e-06, respectively. However, because 26 and 21 amino acid residues, respectively, aligned to the Dof domain were too short to act as a functional zinc-finger Dof domain, these two genes were excluded. Detail information regarding the *SmeDof* genes, including name, coding protein, CDS length, molecular weight and PI value, is shown in Table 1.

## Phylogenetic relationship of the DOF proteins in major plant species

The formation of the plant-specific Dof transcription factors is associated with the evolutionary process of plants (*Taylor & Raes, 2004*; *Shigyo et al., 2007*). To provide insight into the evolution of the Dof family of genes, we performed phylogenetic analysis across representative plant species, including an alga, moss, lycophyte, gymnosperm, eudicot and monocot. Detail information of the gene family in different plant species is shown in Table S1. However, the lycophyte Dof proteins were removed from the final phylogenetic tree because of the lack of complete and functional domains. A phylogenetic tree is usually constructed by a neighbour-joining or maximum-likelihood method using sequences of whole-length proteins, domains or CDS regions. In this study, the neighbour-joining method with 1,000 bootstraps using the Dof domain sequence was performed to construct the phylogenetic tree (Fig. 1), which was generally consistent with previous reports (*Shigyo et al., 2007*; *Moreno-Risueno et al., 2007*; *Ma et al., 2015*).

As demonstrated previously, the number of Dof transcription factors varies among land plants (*Yang, Tuskan & Cheng, 2006*), and the results of this study confirm that conclusion. As shown in Fig. 1, the green algae *C. reinhardtii* only had one *Dof* gene in subgroup D1, indicating that *Dof* members in other groups might originate from subgroup D1 members;
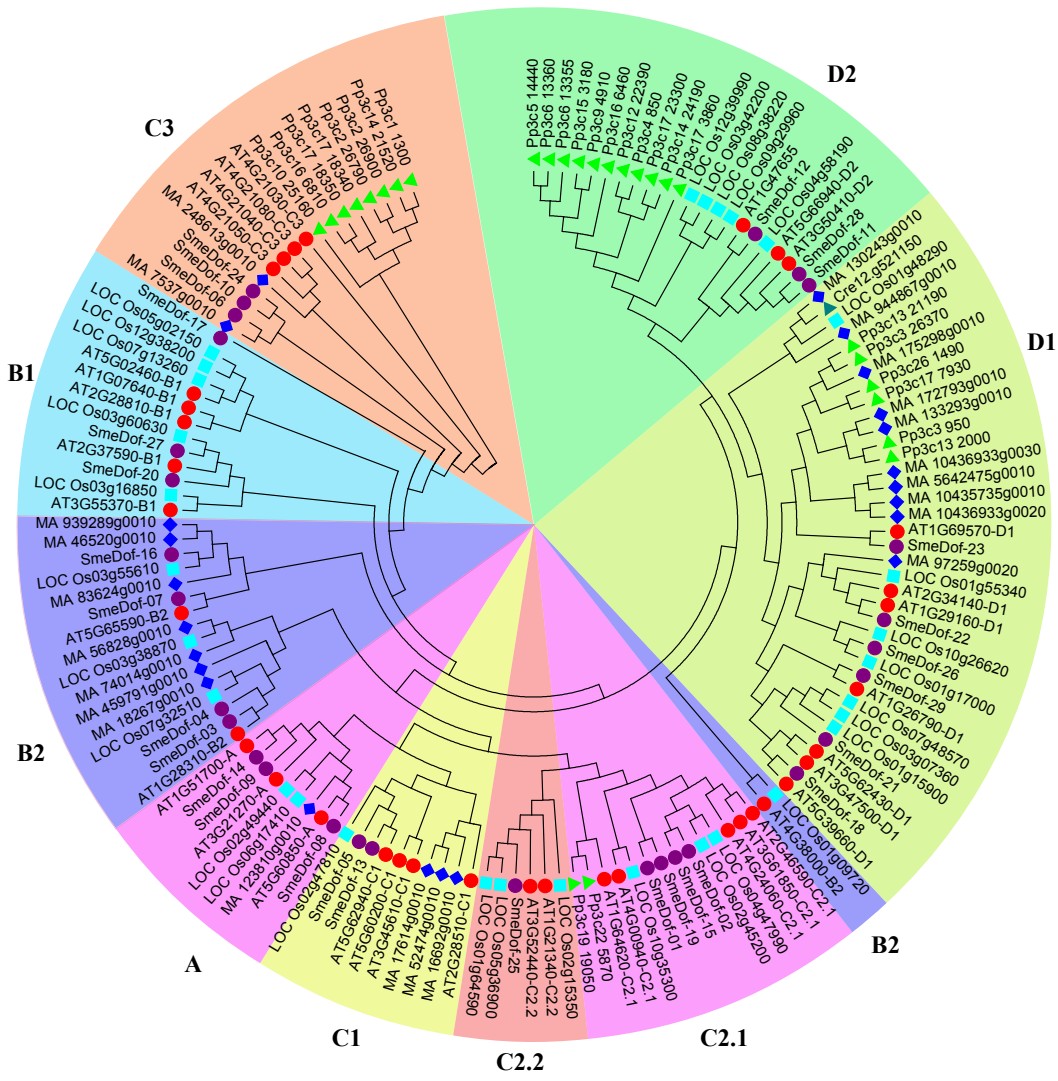

**Figure 1** **The phylogenetic tree of the *Dof* genes from six plant species.** Individual species are distinguished by different shapes with different colours. The prefixes of Cre, Pp3c, MA, LOC_Os, AT and SmeDof indicate these genes were found in *C. reinhardtii*, *P. patens*, Norway spruce, rice, *Arabidopsis* and eggplant, respectively.

whereas the number of Dof transcription factors in the moss *P. patens* enlarged into 27 genes. In *P. patens*, 11, 8, 6 and 2 *Dof* genes were classified into subgroups C3, D1, D2 and C2.2, respectively; thus, these four clades were likely relatively original compared with other clades. In some species, the *Dof* genes are enriched in certain branches. Multiple *Dof* genes of *P. patens* in subgroups C3 and D1 were aggregated into small branches that split with *Dof* genes in the other species, which possibly resulted from *Dof* gene duplication in *P. patens*. A total of 23 *Dof* genes were found in *Picea abies* distributed in all subgroups. According to the phylogenetic tree, the 29 eggplant *Dof* genes were distributed in all 9 branches, with 3, 2, 4, 2, 4, 2, 1, 6 and 3 in subgroups A, B1, B2, C1, C2.1, C2.2,

**Table 1  Classification and characterization of eggplant *Dof* genes.**

| SmeDof name | Protein name | Group | CDS length | Whole gene length | Molecular weight | pI | Instability index | Instability status | Aliphatic | GRAVY |
|---|---|---|---|---|---|---|---|---|---|---|
| SmeDof-01 | *Sme2.5_02793.1_g00001.1* | C2.1 | 4,037 | 302 | 33,233.76 | 8.46 | 47.87 | unstable | 60 | −0.766 |
| SmeDof-02 | *Sme2.5_00047.1_g00017.1* | C2.1 | 2,071 | 281 | 30,200.42 | 9.03 | 33.68 | stable | 59.68 | −0.622 |
| SmeDof-03 | *Sme2.5_00720.1_g00002.1* | B2 | 3,408 | 391 | 42,895.37 | 6.98 | 52.72 | unstable | 62.58 | −0.672 |
| SmeDof-04 | *Sme2.5_02054.1_g00006.1* | B2 | 1,242 | 264 | 29,975.35 | 8.16 | 41.07 | unstable | 64.92 | −0.746 |
| SmeDof-05 | *Sme2.5_00536.1_g00017.1* | C1 | 1,576 | 288 | 31,920.38 | 6.3 | 41 | unstable | 52.08 | −0.666 |
| SmeDof-06 | *Sme2.5_00019.1_g00022.1* | C3 | 18,793 | 589 | 64,449.89 | 6.14 | 52.01 | unstable | 82.61 | −0.337 |
| SmeDof-07 | *Sme2.5_02584.1_g00003.1* | B2 | 3,886 | 360 | 39,038.66 | 7.09 | 48.45 | unstable | 52.31 | −0.736 |
| SmeDof-08 | *Sme2.5_00326.1_g00010.1* | A | 7,425 | 386 | 41,284.63 | 5.94 | 44.67 | unstable | 58.16 | −0.581 |
| SmeDof-09 | *Sme2.5_01232.1_g00017.1* | A | 1,629 | 182 | 19,981.01 | 8.14 | 56.25 | unstable | 53.13 | −0.815 |
| SmeDof-10 | *Sme2.5_03054.1_g00002.1* | C3 | 2,634 | 293 | 32,721.61 | 7.48 | 48.1 | unstable | 59.25 | −0.661 |
| SmeDof-11 | *Sme2.5_01191.1_g00008.1* | D2 | 6,024 | 243 | 25,553.48 | 8.72 | 46.98 | unstable | 57.08 | −0.423 |
| SmeDof-12 | *Sme2.5_18208.1_g00003.1* | D2 | 1,061 | 232 | 24,021.45 | 6.58 | 49.43 | unstable | 51.29 | −0.582 |
| SmeDof-13 | *Sme2.5_00151.1_g00002.1* | C1 | 2,298 | 297 | 32,940.49 | 6.75 | 39.71 | stable | 57.1 | −0.718 |
| SmeDof-14 | *Sme2.5_01155.1_g00012.1* | A | 1,301 | 217 | 23,346.49 | 9.17 | 53.57 | unstable | 44.06 | −0.872 |
| SmeDof-15 | *Sme2.5_03699.1_g00001.1* | C2.1 | 2,569 | 267 | 29,110.24 | 9.21 | 48.03 | unstable | 56.97 | −0.676 |
| SmeDof-16 | *Sme2.5_01191.1_g00002.1* | B2 | 1,185 | 290 | 31,399.01 | 9.03 | 50.92 | unstable | 65 | −0.563 |
| SmeDof-17 | *Sme2.5_04366.1_g00001.1* | B1 | 9,807 | 261 | 28,334.34 | 9.4 | 35.24 | stable | 53.1 | −0.713 |
| SmeDof-18 | *Sme2.5_01247.1_g00005.1* | D1 | 1,913 | 371 | 41,024.83 | 7.59 | 37.61 | stable | 66.98 | −0.725 |
| SmeDof-19 | *Sme2.5_00013.1_g00005.1* | C2.1 | 1,607 | 217 | 24,352.14 | 6.98 | 54.26 | unstable | 63.73 | −0.721 |
| SmeDof-20 | *Sme2.5_00146.1_g00017.1* | B1 | 1,302 | 301 | 33,581.31 | 9.48 | 61.73 | unstable | 55.35 | −0.744 |
| SmeDof-21 | *Sme2.5_00669.1_g00003.1* | D1 | 2,744 | 504 | 54,727.03 | 5.66 | 52.16 | unstable | 56.31 | −0.787 |
| SmeDof-22 | *Sme2.5_02533.1_g00002.1* | D1 | 812 | 166 | 18,642.8 | 8.87 | 40.68 | unstable | 52.89 | −0.873 |
| SmeDof-23 | *Sme2.5_03125.1_g00004.1* | D1 | 4,976 | 445 | 49,545.73 | 6.47 | 48.04 | unstable | 54.36 | −0.859 |
| SmeDof-24 | *Sme2.5_00720.1_g00003.1* | C3 | 1,335 | 305 | 33,223.53 | 9.43 | 70.52 | unstable | 44.46 | −0.913 |
| SmeDof-25 | *Sme2.5_09080.1_g00001.1* | C2.2 | 1,225 | 258 | 29,384.18 | 4.55 | 50.9 | unstable | 53.33 | −0.825 |
| SmeDof-26 | *Sme2.5_00161.1_g00001.1* | D1 | 4,032 | 467 | 52,088.65 | 5.9 | 43.24 | unstable | 51.58 | −0.846 |
| SmeDof-27 | *Sme2.5_02556.1_g00009.1* | B1 | 1,679 | 370 | 39,849.03 | 8.98 | 63.55 | unstable | 52.22 | −0.652 |
| SmeDof-28 | *Sme2.5_08918.1_g00002.1* | D2 | 920 | 205 | 22,056.75 | 9.23 | 55.35 | unstable | 62.34 | −0.42 |
| SmeDof-29 | *Sme2.5_00298.1_g00013.1* | D1 | 2,953 | 525 | 57,189.32 | 6.01 | 56.78 | unstable | 56.32 | −0.835 |

C3, D1 and D2, respectively (Fig. 1). In algae, only one Dof factor was identified in the green alga *C. reinhardtii* and none in the red alga *Cyanidioschyzon merolae* (*Yanagisawa, 2002*; *Yanagisawa, 2004*; *Shigyo et al., 2007*). By contrast, *Glycine max* contains the high number of 93 members of Dof family genes (*Ma et al., 2015*). These results revealed that the number of *Dof* genes increases from algae to higher plants, suggesting an expansion of Dof family transcription factors during the evolution of lower plants to terrestrial plants. This conclusion is consistent with previous study on Dof TFs in Chinese cabbage (*Ma et al., 2015*). Moreover, the origin of the Dof transcription factors predates the divergence between the green algae and the ancestors of land plants (*Moreno-Risueno et al., 2007*; *Shigyo et al., 2007*).
The *Dof* genes in rice were found in 9 branches, but were absent in subgroup C3, which was similar to cucumber (*Wen et al., 2016*). The *Dof* members in subgroup C3 originated after moss and were more divergent than members in other groups, although none were identified in rice. Three members of eggplant *SmeDof* genes were classified into subgroup C3; however, the domain sequences were also disparate with members in *Arabidopsis*. Gene duplication in a single species is a possible result from genome duplication after the divergence between asterids and eurosides II clades, which were the ancestors of *Arabidopsis* and eggplant.

## Phylogenetic relationships of the Dof proteins among Solanaceae species

We constructed a phylogenetic tree of the *Dof* genes from six species of Solanaceae based on the Dof domains using the neighbour-joining method (Fig. 2, Table S1). As shown in Table 2, five of the six species, *S. melongena, S. lycopersicum, S. tuberosum, C . annuum* and *P. axillaris*, had similar numbers of *Dof* genes ranging from 29 to 33. By contrast, the allotetraploid *Nicotiana tabacum* had 60 Dof TF coding genes, which was likely because of genome duplication events increasing gene copies. The Dof proteins in tobacco were enriched in 3 subgroups: B1, C2.1, and D1, with 13, 16, and 14 genes, respectively. However, the *Dof* genes of *N. tabacum* in subgroup C2.2 did not differ much from those of the other five species, except for those of petunia, which had 4 genes in this group. The Dof TFs of eggplant, tomato, potato and pepper in subgroups C1 and D1 were the same, with 2 and 6, respectively. Except for the allotetraploid tobacco, *Dof* genes from different species varied slightly from A to D2 subgroups, indicating an even distribution within the family Solanaceae.

The genome size of eggplant (833.1 Mb, 42,035 coding genes; *Hirakawa et al., 2014*), tomato (950 Mb, 34,727 coding genes; *Tomato Genome Consortium, 2012*) and potato (844 Mb, 35,119 coding genes; *Potato Genome Sequencing Consortium, 2011*) did not vary significantly, which corresponded to their similar number of Dof transcription factor encoding genes, with 29, 33, and 32 members, respectively. Notably, although the genome size of hot pepper (3.48 Gb, 34,899 coding genes; *Kim et al., 2014*) was approximately fourfold larger than that of its close relatives tomato and eggplant, the total coding genes and *Dof* genes did not have proportionally more. We propose that because hot pepper is a diploid species which did not experience the recent whole-genome duplication (*Kim et al., 2014*), and therefore, no significant enlargement of *Dof* genes occurred. The genome size of another solanaceous species *N. tabacum* (4.41–4.57 Gb, 85,439 coding genes; *Sierro et al., 2014*) was almost fivefold larger than that of eggplant, but the number of Dof transcription factors (60 members) was fewer than twice that of eggplant (29 members), which might be attributed to gene loss after the allopolyploidization event.

## Gene structure and conserved motifs analyses of the eggplant *Dof* genes

The exon-intron organization can be used as supporting evidence to determine the evolutionary relationships among genes or organisms  (*Koralewski & Krutovsky, 2011*; *Malviya et al., 2015*). Therefore, structural analysis of gene families with phylogenetic

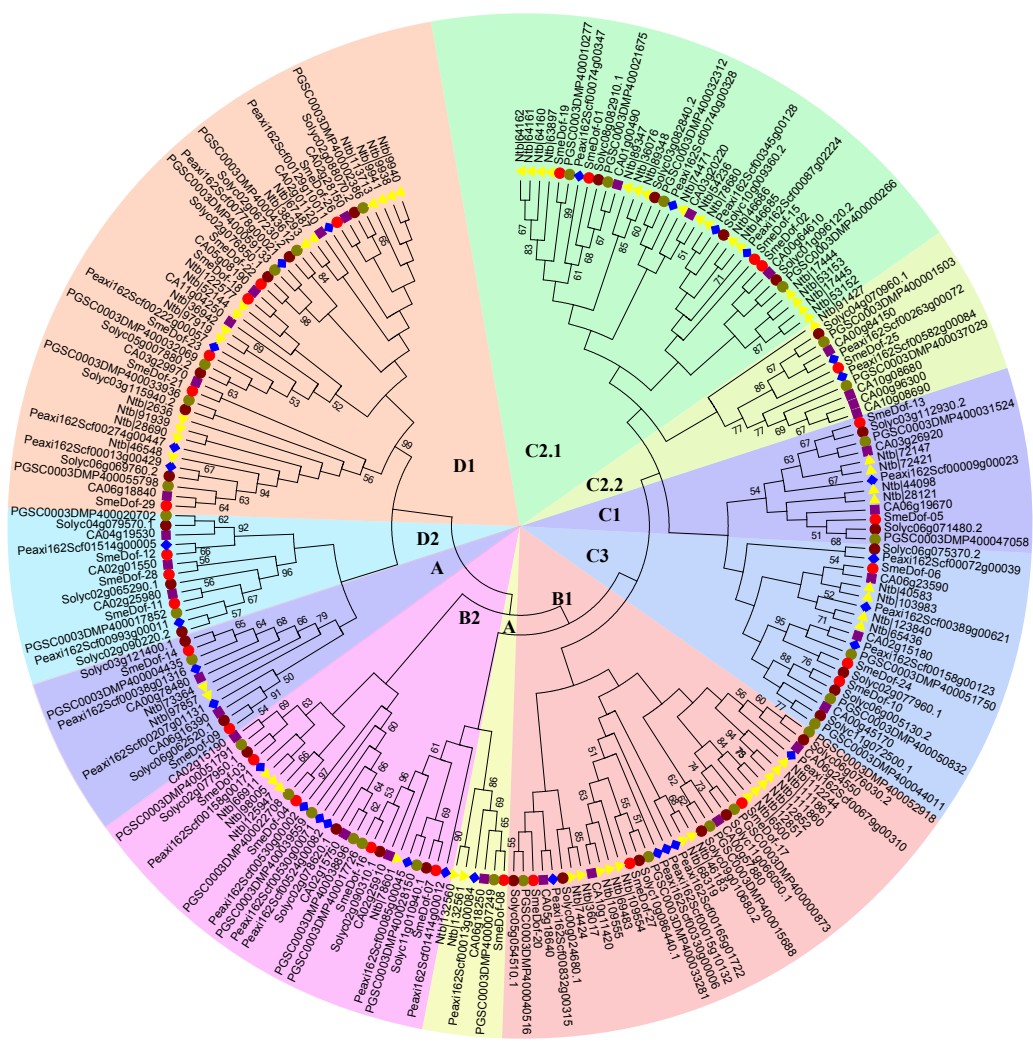

**Figure 2** **The phylogenetic tree of the *Dof* genes from six plant species in Solanaceae.** Individual species were distinguished by different shapes with different colors. Bootstrapping values are indicated as percentages (when >50%) along the branches. The prefix of SmeDof, Solyc, PGSC, Peaxi, CA and Ntb indicated these genes were found in eggplant, tomato, potato, petunia, pepper and tobacco.

**Table 2** **Detailed information on the Dof members in each subgroup of the six plant species from family Solanaceae.**

| Plant species of *Solanaceae* | Subgroup | | | | | | | | | Unclassified | Total |
|---|---|---|---|---|---|---|---|---|---|---|---|
| | A | B1 | B2 | C1 | C2.1 | C2.2 | C3 | D1 | D2 | | |
| *Solanum melongena* | 3 | 3 | 4 | 2 | 4 | 1 | 3 | 6 | 3 | 0 | 29 |
| *Solanum lycopersicum* | 2 | 6 | 4 | 2 | 4 | 1 | 4 | 6 | 3 | 1 | 33 |
| *Solanum tuberosum* | 2 | 5 | 6 | 2 | 4 | 2 | 3 | 6 | 2 | 0 | 32 |
| *Capsicum annuum* | 3 | 4 | 3 | 2 | 3 | 4 | 3 | 6 | 3 | 2 | 33 |
| *Petunia axillaris* | 3 | 5 | 6 | 1 | 4 | 1 | 3 | 5 | 2 | 3 | 33 |
| *Nicotiana tabacum* | 4 | 13 | 4 | 4 | 16 | 1 | 4 | 14 | 0 | 0 | 60 |

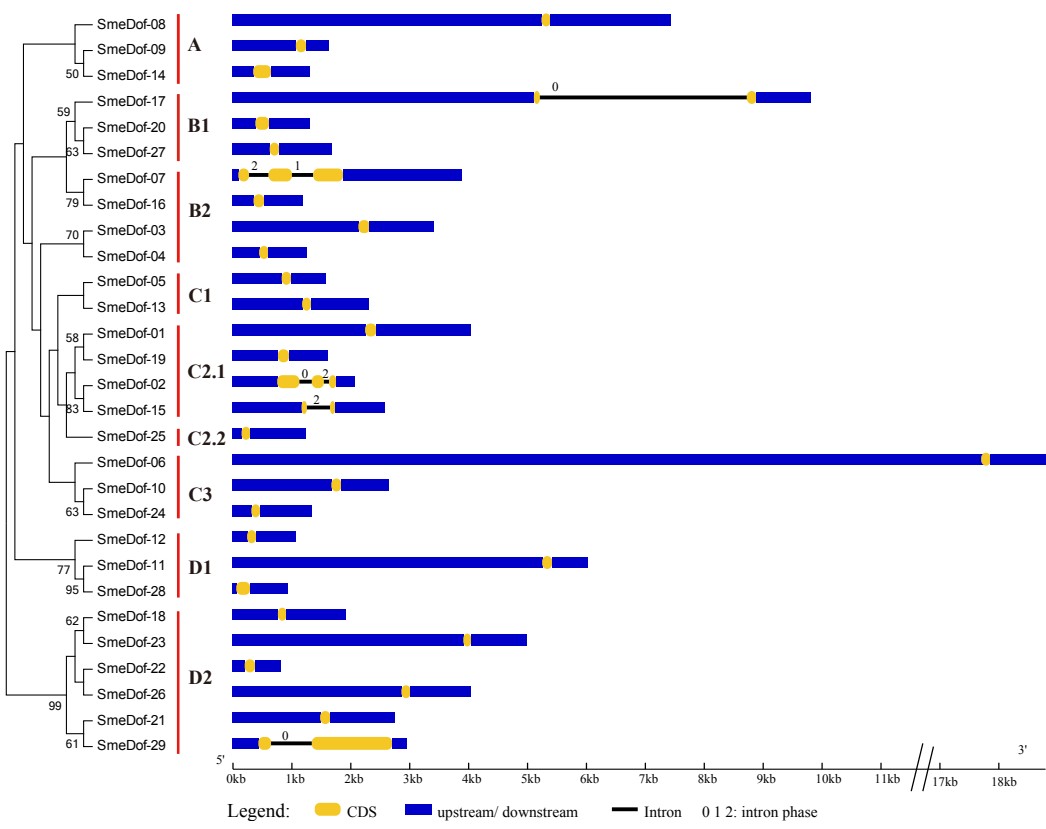

**Figure 3  Gene structure of the eggplant *Dof* genes.** Yellow boxes represent CDS, and black and blue lines represent introns and UTRs, respectively. The lengths of the exons, introns and UTRs were drawn according to the length of sequences.

relationships can provide valuable information on duplication events and evolutionary patterns (*Cai et al., 2013*). To investigate the mechanisms of structural evolution of *SmeDof* genes in eggplant, we compared the exon-intron structures of 29 annotated *SmeDof* genes. A detailed illustration of gene structure including the CDS, intron and UTR structures is shown in Fig. 3. Similar to the structure of *Arabidopsis*, rice, cucumber and tomato *Dof* genes (*Lijavetzky, Carbonero & Vicente-Carbajosa, 2003*; *Wen et al., 2016*; *Cai et al., 2013*), the eggplant *Dof* genes had few introns, which ranged in number from 0 to 2 in each gene (Fig. 3). Three of the *SmeDof* genes, *SmeDof-15*, *SmeDof-17* and *SmeDof-29*, had one intron, whereas *SmeDof-02* and *SmeDof-07* both had two introns. Moreover, 24 of 29 *SmeDof* genes were found without introns. *SmeDof* genes with introns showed either "0", "1", or "2" intron phases and were in four different subgroups: B1, B2, C2.1, and D2. The gene *SmeDof-17* had the longest intron. Of all eggplant *Dof* genes, *SmeDof-06* was the longest gene with the longest UTRs, whereas the shortest gene was *SmeDof-22*.

The diversity of motif compositions in SmeDof proteins was assessed using the MEME programme, and a total of 15 conserved motifs were identified. The schematic distribution of these 15 motifs among SmeDof proteins is shown in Figs. 4 and 5. The motif, motif1, was uniformly observed in all SmeDof proteins and was confirmed as the conserved Dof domain.

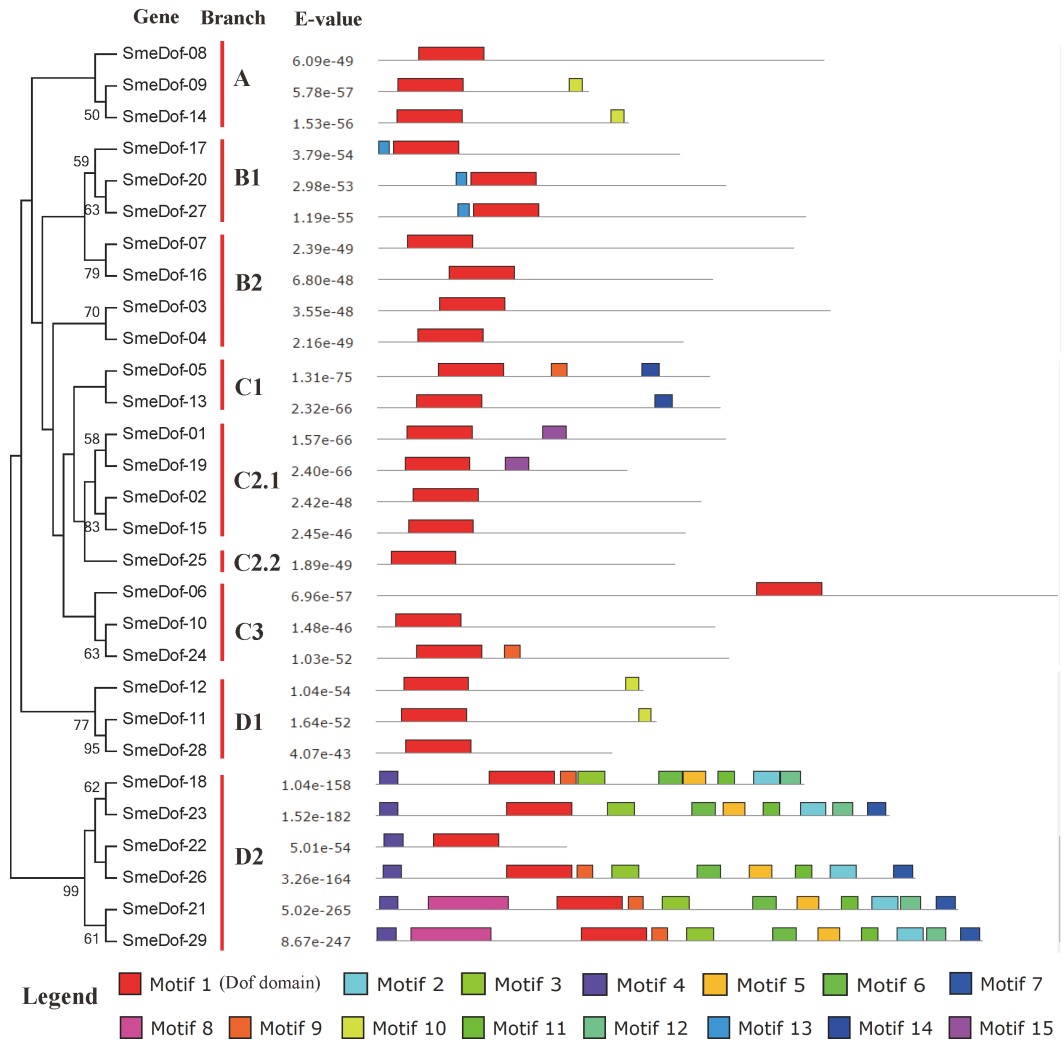

**Figure 4** Conserved motifs embedded in the eggplant *Dof* genes.

Similar to the results from *Arabidopsis*, rice, cucumber and tomato (*Lijavetzky, Carbonero & Vicente-Carbajosa, 2003*; *Wen et al., 2016*; *Cai et al., 2013*), this result suggested that SmeDof transcription factors were evolutionarily conserved in plants. Groups B2 and C2.2 and six other genes (*SmeDof-02*, *SmeDof-06*, *SmeDof-08*, *SmeDof-10*, *SmeDof-15*, and *SmeDof-28*) had only one conserved motif1 (Fig. 4). All members in group A contained motif10 except for *SmeDof-08*, whereas motif13 was in all SmeDof proteins in Group B1. Both Dof proteins in Group C1 contained motif1 and motif14, with *SmeDof-05* containing the additional motif9. Members of group D2 generally had more than 9 motifs except for *SmeDof-22*, which only contained motif1 and motif4; all other members had motif2, motif3, motif4, motif5, motif6, and motif11. The *SmeDof* genes in each subgroup had several unique motifs, suggesting that Dof proteins within the same subclusters shared certain functional similarities. Distribution of the motifs revealed that the *SmeDof* genes were likely conserved during evolution.

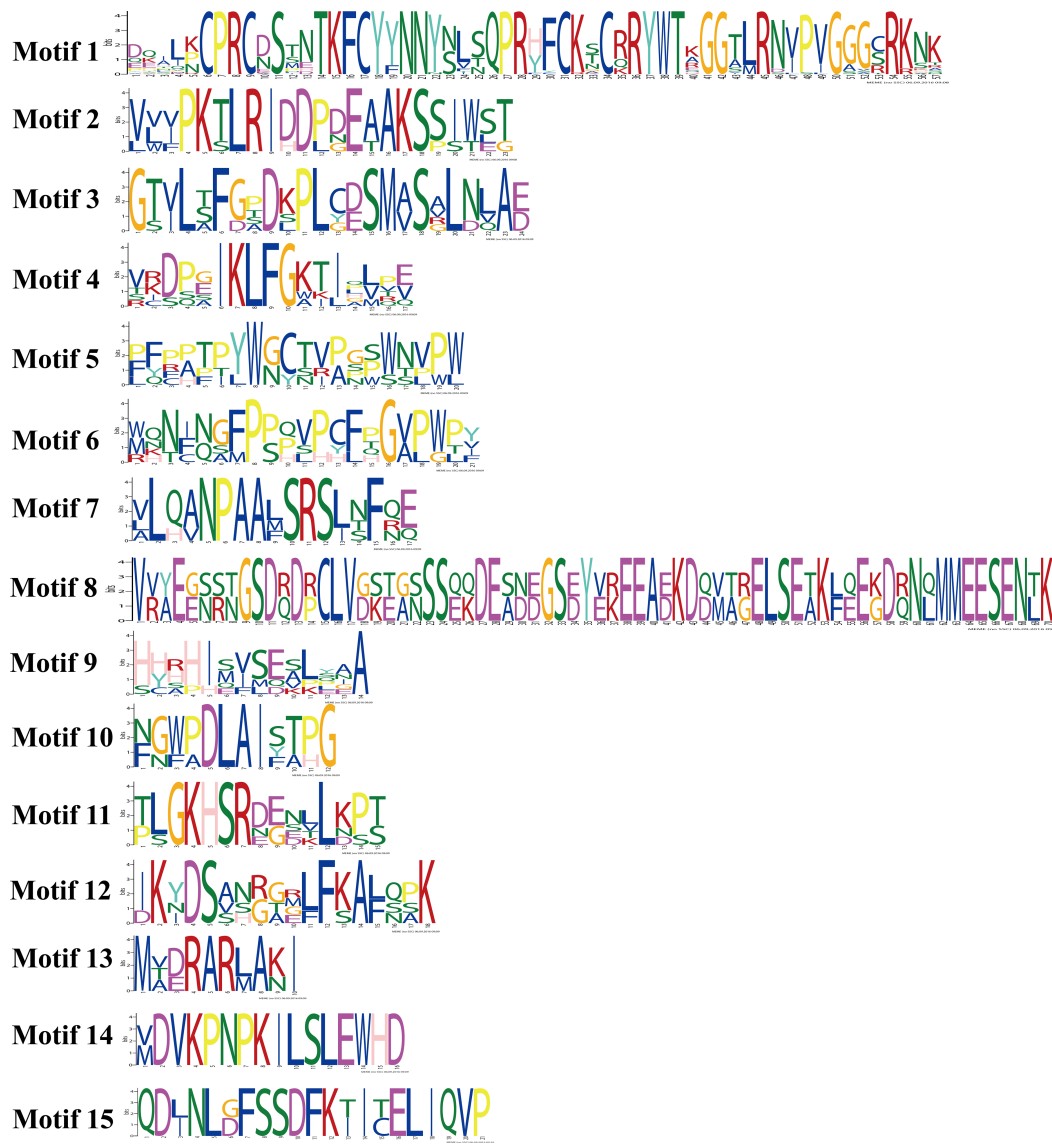

**Figure 5 Conserved motifs embedded in the eggplant *Dof* genes.** Amino acid sequences of each motif. The font size represents the frequency of the respective amino acid.

## Identification of orthologous, co-orthologous and paralogous *Dof* genes in eggplant, tomato, *Arabidopsis* and rice

The relationships of orthologous, co-orthologous, and paralogous *Dof* genes among four species were investigated. Orthologs are genes evolved by vertical descent from a single ancestral gene whereas paralogs are originated by duplication. The co-orthologs are subdivided type of orthologs, which are two or more genes in one lineage that are, collectively, orthologous to one or more genes in another lineage due to a lineage-specific duplication(s) (*Koonin, 2005*). The comparative analysis was performed using OrthoMCL with default settings, and the results are presented in Fig. 6 and Table S2. Among the 29
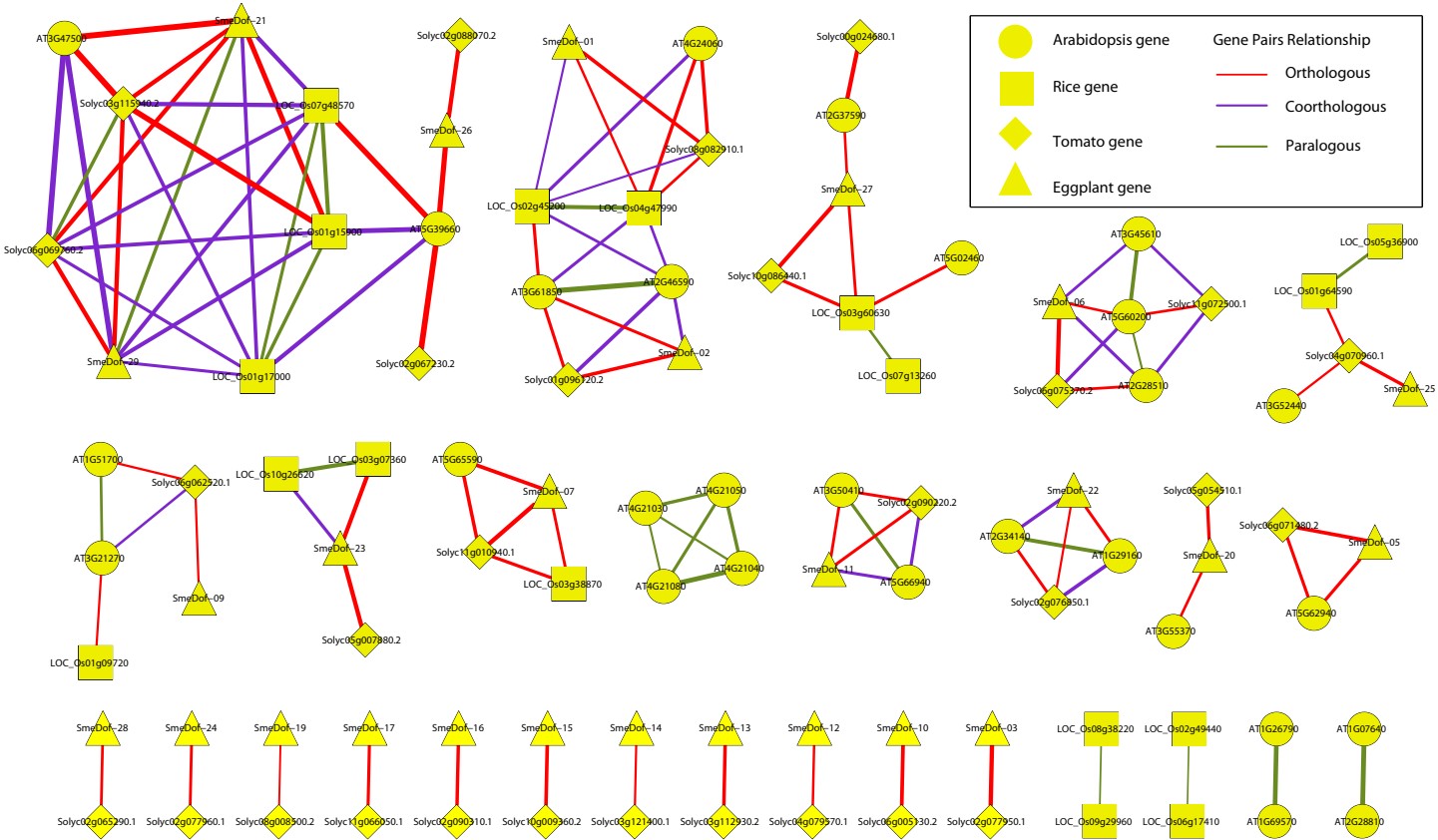

**Figure 6** **Ortholog, co-ortholog and paralog gene pairs.** The ellipse, rectangle, triangle and diamond indicate that a gene belongs to *A. thaliana*, *O. sativa*, *S. lycopersicum* and *S. melongena*, respectively, The red, purple and green lines indicate orthologous, co-orthologous and paralogous relationships, respectively, and the width of the line is associated with the relationship index produced by OrthoMCL software.

*SmeDof* genes of eggplant and the 33 *Dof* genes of tomato, 28 orthologous gene pairs were identified. Ten orthologous gene pairs were identified among the 29 *SmeDof* genes and the 36 *Dof* genes of *Arabidopsis*, whereas only five orthologous gene pairs were found between eggplant and rice (30 *Dof* genes). The most orthologous genes were between eggplant and tomato, which could be explained because both species are in the family Solanaceae. We also identified six co-orthologous gene pairs between eggplant and *Arabidopsis* and seven pairs between eggplant and rice. Paralogous genes were detected in all four species, with 14 pairs in *Arabidopsis*, nine pairs in rice, one pair in tomato, and one gene pair in eggplant. The genes *SmeDof-21* and *SmeDof-29* were the only paralogous *Dof* genes in eggplant.

## Expression pattern of the *Dof* genes among different eggplant varieties and close species

To increase our understanding of the expression profiles of the *Dof* among different eggplant varieties and close species, we investigated gene expression for each *SmeDof* gene using RNA-seq data of six samples: two cultivated eggplants, three closest wild relatives and a semi-wild relative. The FPKM expression of *SmeDof* genes in six eggplants is provided

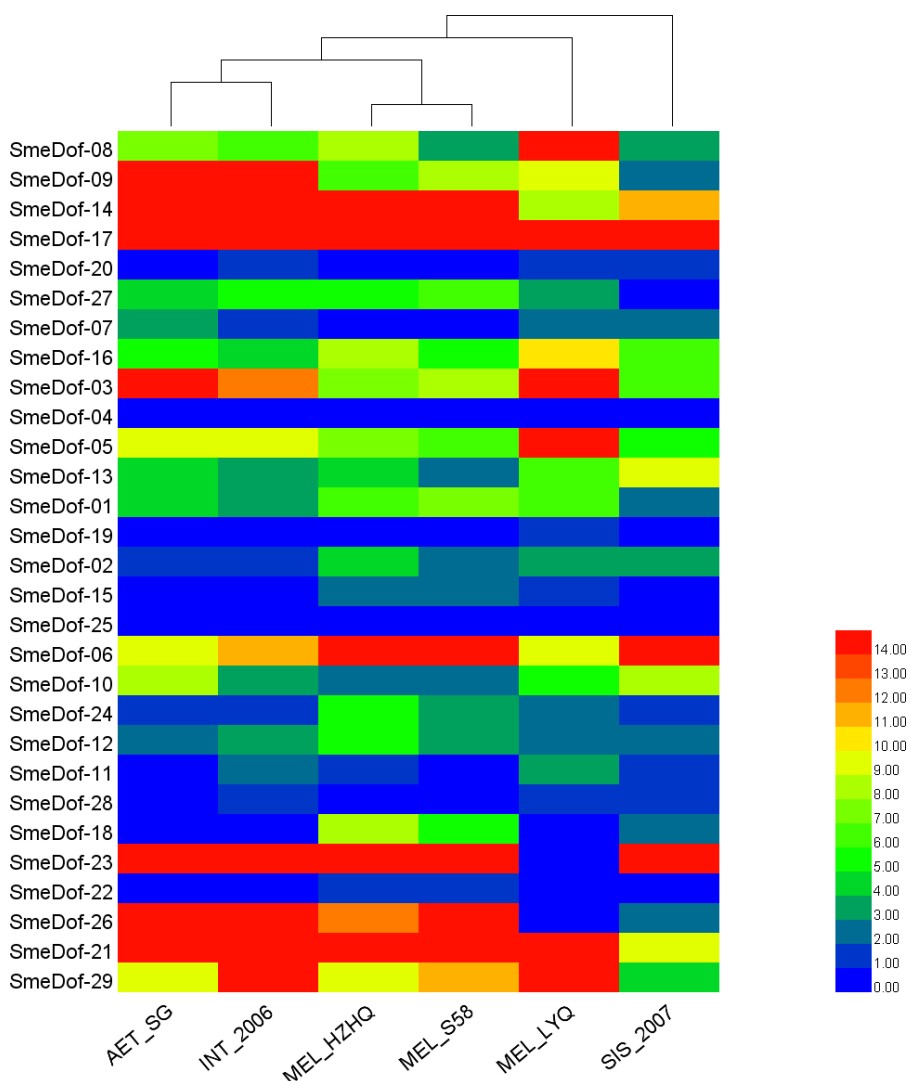

**Figure 7** **The heat map shows the expression profile of the eggplant *Dof* genes in six eggplants.** The *x*-axis represents the names of the six eggplants, and the *y*-axis represents different *SmeDof* genes. The expression levels of SmeDof genes are revealed by different colours, which increase from blue to red.

in Table S3. As shown in Table 3 and Fig. 7, the wild species AET_SG and INT_2006 were closely related. The alignment rate between RNA-seq reads of the wild SIS_2007 and the published eggplant genome sequence was only 13.40%, and therefore, this eggplant was the most distant from domesticated eggplants (Fig. 7). The cultivated eggplant MEL_HZHQ with linear, long fruits was phylogenetically more closely related to the semi-wild species MEL_S58 instead of the other cultivated type MEL_LYQ, which has round fruits. Moreover, MEL_HZHQ is similar to the semi-wild species in morphology, with vine branches and multi-pistillate flowering. The low expression values of genes in SIS_2007, AET_SG, and INT_2006 were highly supported by that these plant materials were close wild relatives to eggplant, not varieties of eggplant. The left reads mapped ratio were 70%–88% and the

**Table 3  The alignment rates of RNA-seq sequences of six eggplants against the eggplant reference genome.**

| Sample name | Subspecies | Eggplant type | Left reads mapped | Left multi mapped | Concordant mapped |
|---|---|---|---|---|---|
| AET_SG | *S. aethiopicum gr. Gilo* | wild | 70.20% | 4.70% | 56.40% |
| INT_2006 | *S.intergrifolium* | wild | 71.80% | 5.00% | 58.40% |
| MEL_HZHQ | *S. melongena* | cultivated | 87.40% | 4.70% | 81.70% |
| MEL_LYQ | *S. melongena* | cultivated | 88.60% | 5.00% | 82.00% |
| MEL_S58 | *S. melongena* | semi-wild | 86.90% | 5.00% | 80.60% |
| SIS_2007 | *S. sisymbriifolium* | wild | 26.70% | 6.30% | 13.40% |

concordant mapped ratio were 56%–82% except SIS_2007. The mapped ratio were in a normal range of RNA-Seq mapping ratio, as most studies have a concordant mapped ratio were 50%–90%.

The expression levels of different *SmeDof* genes in the six types of eggplants are represented by different colours and are shown in Fig. 7. *SmeDof-25* and *SmeDof-04* did not express in any of the eggplants. The expression of *SmeDof* genes *SmeDof-20*, *SmeDof-19*, and *SmeDof-22* was either very low or not detecable in the six types of eggplants, whereas *SmeDof-17* was highly expressed in all types of eggplants. The expression patterns of *SmeDof-05*, *SmeDof-08*, and *SmeDof-23* in MEL_LYQ were different from those in the other five eggplants; *SmeDof-05* and *SmeDof-08* were highly expressed in MEL_LYQ, but the expression was comparatively low in other eggplants. By contrast, *SmeDof-23* had high expression levels in the other five eggplants but did not express in MEL_LYQ. The two genes *SmeDof-21* and *SmeDof-27* had different expression in the wild eggplant SIS_2007 compared with that in the other types of eggplants. The expression levels of *SmeDof-29* varied significantly between AET_SG and INT_2006. However, the functions of *Dof* genes in different ecotypes of eggplants remain to be further investigated.

## CONCLUSIONS

We conducted a comprehensive analysis of Dof transpcrition factors in the genome of eggplant. A total of 29 *SmeDof* genes encoding Dof transpcrition factors were identified from eggplant, which were classified into four groups and nine subgroups: A, B1, B2, C1, C2.1, C2.2, C3, D1 and D2. The gene structure, conserved motifs and homologous genes of *Dof* genes in eggplant were investigated. All of the SmeDof proteins contained motif1 which was considered as the conserved Dof domain. A comparative study of the Dof family factors between eggplant and other plant species was performed that could facilitate the functional analysis of Dof factors in eggplant. Moreover, we also investigated the expression patterns of *Dof* genes in different types of eggplants. Our study provides valuble information for further understanding of the mechanisms underlying abiotic stress response of SmeDof transcription factors.

## ACKNOWLEDGEMENTS

We thank Nanjing Huasequen Biotechnologies Co, Ltd., China for assistance on bioinformatics analysis.

### Funding

This work was supported by the Natural Science Foundation of Zhejiang (LQ18C150004), and the New Variety Breeding Project of the Major Science and Technology Projects of Zhejiang (2016C02051-2). The funders had no role in study design, data collection and analysis, decision to publish, or preparation of the manuscript.

### Grant Disclosures

The following grant information was disclosed by the authors:
Natural Science Foundation of Zhejiang: LQ18C150004.
Major Science and Technology Projects of Zhejiang: 2016C02051-2.

### Competing Interests

The authors declare there are no competing interests.

### Author Contributions

- Qingzhen Wei conceived and designed the experiments, performed the experiments, analyzed the data, prepared figures and/or tables, authored or reviewed drafts of the paper, approved the final draft.
- Wuhong Wang analyzed the data, authored or reviewed drafts of the paper, approved the final draft.
- Tianhua Hu, Haijiao Hu, Weihai Mao and Qinmei Zhu analyzed the data, contributed reagents/materials/analysis tools, authored or reviewed drafts of the paper, approved the final draft.
- Chonglai Bao conceived and designed the experiments, authored or reviewed drafts of the paper, approved the final draft.

### Data Availability

The raw data of RNA-seq experiment is now provided and deposited in Sequence Read Archive (NCBI): SRP127743.

### Supplemental Information

Supplemental information for this article can be found online at http://dx.doi.org/10.7717/peerj.4481#supplemental-information.

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
