# Peer review of "Genome-wide identification and characterization of Dof transcription factors in eggplant (Solanum melongena L.)"

_PeerJ, doi:10.7717/peerj.4481_

## Round 0.1 · original submission · Major Revisions

· Academic Editor

Major Revisions

There were major concerns with the manuscript in that there was really no new data, except an evaluation based on unavailable data. There would be a need to see or benchmark new data against the observations proposed, and to observe the sampling and methods used. A more descriptive form of the manuscript is required to justify rationale for the work and methods used along with some evidence of new data to form the resulting discussion. Please consider the recommendations provided by the reviewers. Perhaps in the resubmission process some of the issues raised will be addressed. Additional data from new species is of interest for assessing the range of function for gene families and this would be included granted that readership would be comfortable accepting the new data provided.

Reviewer 1 ·

Basic reporting

This manuscript reports the Dof family identified in the genome sequence data of eggplant, Solanum melongena. This paper is too descriptive and new suggestions and insights into genetics and physiology of eggplant are rather limited. Most of all analysis is based on published data. Even though the authors performed RNA-Seq analysis across six varieties, the authors do not disclose raw sequence data, which is conflict to the guidline of submission instruction of PeerJ.
https://peerj.com/about/author-instructions/#data-and-materials

Experimental design

Details RNA-Seq analysis are required in Materials and methods section. What organs and stages, how many replicates, and what types of sequencers did the authors employ? How much data for each sample?

Validity of the findings

The concordant map rates in wild relatives, SIS_2007, AET_SG, and INT_2006 were low. Dose this result reflect gene expression profile correctly? DNA polymorphisms can disturb read mapping on the reference sequences of cultivated eggplant, resulting in underestimation of the expression.

Additional comments

The authors performed phylogenetic analysis, gene structural analysis, ortholog/paralog identification, and expression analysis separately. The results should be compared each other to evaluate that the subfamilies are evolved structurally and functinally parallel or not. How about regulatory sequences, like promoters?

Reviewer 2 ·

Basic reporting

No comment.

Experimental design

1. For in planta classification of eggplant SmeDof genes into different groups/subgroups such as A, B, C D etc, the authors have chosen 4 additional species, namely C. reinhardtii, P. patens, A. thaliana, O. sativa. However, the rationale behind this choice is not provided in sufficient detail. Thus, it is not clear if the classification results would change if additional or different species were included in the analysis. For example, the authors cite Moreno-Risueno et al., 2007, whose analysis indicates the presence of a subgroup G but the representative protein sequences from that sub-group are not included in the current manuscript. It is suggested that to maintain consistency with previously published results, the authors include representative sequences from all subgroups in their phylogenetic analysis. One way to achieve this would be to include all the species that have been used in previously published classifications. Ideally, this would include the species at least from the papers that the authors have cited, namely Cai XF et al. 2013, Guo and Qiu 2013, Huang et al 2016, Kushwaha et al. 2011, Lijavetzky et al 2003, Ma et al. 2015, Malviya et al 2015, Moreno-Risueno et al. 2007 and Shigyo et al. 2007 etc. Alternatively, the authors should better justify their choice of species and protein sequences.

2. What plant tissue was used as the source of RNA for the RNA-Seq experiments? This needs to be clearly mentioned, otherwise the statements such as in line 306 “SmeDof-23 had high expression levels in the other five eggplants but did not express in MEL_LYQ” do not remain very meaningful.

3. In the methods section, the authors state in lines 140 to 142: “The phylogenetic relationships of the six eggplants were determined by the alignment rate between their RNA-seq reads and the eggplant genome sequence (Hirakawa et al. 2014) using BWA software (Li et al., 2009).” Although sequence alignment rate is indicative of evolutionary relatedness, it is not a metric for inferring phylogenetic relationships. The authors should accordingly replace the phrase “phylogenetic relationships” in the sentence with a more suitable descriptor.

4. In the section “Phylogenetic relationships of the Dof proteins among Solanaceae species”: the authors have tried to correlate the genome size with the number of Dof proteins. However, a more suitable comparison would be with the total number of protein-coding genes in each genome, instead of the raw genome size. This is because even without whole-genome duplications, genome size can increase dramatically due to increase in non-coding repeats, and transposons etc.

Validity of the findings

It is not clear if the classification into different subgroups such as A, B, C D etc is robust to the inclusion of additional plant species. See the issues raised in the “Experimental Design” section above.

Additional comments

Some language/typo suggestions:
1. Line 37: Please change “anti-biotic stress” to “abiotic stress”.
2. Line 124: Shouldn’t “E-value larger than the 0.05” be “E-value smaller than the 0.05”?
3. Line 136: Please change “RNA-Seq transcriptions” to “RNA-Seq transcriptomes” or similar.
4. Lines 239-240: It is not clear what do the authors mean by “attributed to alternative actions”, please rephrase and explain.
5. Line 288 and elsewhere: Instead of “different eggplants”, it is suggested that the phrase “different eggplant subspecies” or similar should be used.

---

## Round 0.2 · Minor Revisions

· Academic Editor

Minor Revisions

The comments from reviewers were effectively rebuffed or addressed. Are there any comments that can be made regarding the companion publication for this work? The current state of the manuscript is close to an acceptable form. Included are a few notes on the read for the revision which should be considered. I did find in a few cases that observations were basically presented as fact rather than as a proposed hypothesis for the observation, either more comprehensive analyses or some statistical significance would need be applied to the study. I was curious if a note might be applied to address low gene expression when compared to close wild relatives; might this also be explained as a potential difference in the metabolic ecosystem? This may be something to try if another experiment is performed, comparing pooled vs. individual samples. The pointer to the SRA resource is also helpful. Please consider the notes proposed and consider this manuscript in a state of Acceptance requiring Minor Revisions. Thank you for the contribution.

Suggested edits and notes:

Example of annotation:
LINE NO.: / PREVIOUS FORM / SUGGESTED FORM / [ADDITIONAL NOTES, NONE [.]]

30 : / . / , and perhaps validate the role of Dof expression during stress. / [.]
37 : / pose / , thus pose / [.]
47 : / / / [Is there any information that cat explain the different recognition?]
84 : / / / [Is there an example where marker-assisted selection has focused of Dof or the like to select stess traits?]
106 : / in Planta / <i>in planta</i> / [italicize, like 'de novo', 'in vitro'.]
141 : / (AET_SG SIS,_2007 and INT_2006) / / [possible comma correction needed?]
149 : / sample.The / sample. The / [space needed.]
218 : / indicating / suggesting / [not formally validated. Additional stats req.]
249 : / This result was most likely because / We propose ... / [not formally validated. Additional stats req.]
254 : / after the allopolyploidizati on event. // [fix spacing issue.]
273 : / 5. Motif1 was uniformly / 5. The motif, motif1, was uniformily / [.]
287 : / orthologous, co-orthologous and paralogous / / [These terms are often misunderstood; perhaps a section warranting a brief explanation would address reader interpretation. Demonstrate that you know what is being sought other than software output.]
Fig 6 : / Orthomcl / OrthoMCL / [.]

---

## Round 0.3 · accepted · Accept

· Academic Editor

Accept

Thank you for addressing the suggested points from prior reviews. The read is much better and may now be considered Accepted. However as noted below there were still a few questionable formatting issues which may be simple to address. The work provides a nice overview of the Dof transcription factors for eggplant and may provide insights to others in the research area about the role of its range of function. Thank you for the contribution.

Example of annotation:
LINE NO.: / PREVIOUS FORM / SUGGESTED FORM / [ADDITIONAL NOTES, NONE [.]]
129: /1e-5/1.0e-05/ [notation style different from elsewhere in manuscript.]
149: /'tophat/ index.shtml'/./ [space withing URL shown; may need space removed.]
152: /'cole- trapnell-lab'/./ [space withing URL shown; may need space removed.]
203: / fig. 1 / Fig. 1 / [.]
223: / but / , but / [ or re-word sentence to make punctuations work.]
225: / none was / none were / [.]
232: / in table 2 / in Table 2 / [.]
263: / fig. 3 / Fig. 3 / [.]
274: / The motif, motif1 / The motif, motif1, / [.]
294: / duplication(s). (Koonin / duplication(s)(Koonin / [ remove extra '.'.]
310: / table 3 and fig. 7 / Table 3 and Fig. 7 / [.]
319: / Concordant mapped / concordant mapped / [.]
339: / Motif1 / motif1 / [is captitalization important here? Consistency.]